# ECS1 and ECS2 suppress polyspermy and the formation of haploid plants by promoting double fertilization

Yanbo Mao[†‡], Thomas Nakel[†], Isil Erbasol Serbes, Saurabh Joshi, Dawit G Tekleyohans[§], Thomas Baum, Rita Groß-Hardt*

University of Bremen, Centre for Biomolecular Interactions, Bremen, Germany

**Abstract** The current pace of crop plant optimization is insufficient to meet future demands and there is an urgent need for novel breeding strategies. It was previously shown that plants tolerate the generation of triparental polyspermy-derived plants and that polyspermy can bypass hybridization barriers. Polyspermy thus has the potential to harness previously incompatible climate-adapted wild varieties for plant breeding. However, factors that influence polyspermy frequencies were not previously known. The endopeptidases ECS1 and ECS2 have been reported to prevent the attraction of supernumerary pollen tubes by cleaving the pollen tube attractant LURE1. Here, we show that these genes have an earlier function that is manifested by incomplete double fertilization in plants defective for both genes. In addition to supernumerary pollen tube attraction, *ecs1 ecs2* mutants exhibit a delay in synergid disintegration, are susceptible to heterofertilization, and segregate haploid plants that lack a paternal genome contribution. Our results thus uncover ECS1 and ECS2 as the first female factors triggering the induction of maternal haploids. Capitalizing on a high-throughput polyspermy assay, we in addition show that the double mutant exhibits an increase in polyspermy frequencies. As both haploid induction and polyspermy are valuable breeding aims, our results open new avenues for accelerated generation of climate-adapted cultivars.

## Editor's evaluation

This important work convincingly shows that the endopeptidases ECS1 and ECS2 repress the formation of polyspermy-derived triparental offspring and haploid induction by promoting double fertilization. While the underlying mechanisms remain to be further elucidated, the data presented in this study represent a valuable foundation for understanding the regulation of offspring genome size. This study will be of particular interest to the large community of scientists who are interested in plant reproduction and breeding.

## Introduction

Sexually reproducing organisms have evolved a plethora of ingenious tools to make gamete fusion a safe event (*Iwao et al., 2020*; *Tekleyohans and Groß-Hardt, 2020*; *Toda and Okamoto, 2020*). The production of supernumerary sperm is commonly found in both animals and plants, but this strategy comes with the risk of polyspermy. While polyspermy is lethal in most animals, molecular and genetic evidence for supernumerary sperm fusion in plant egg cells has long been lacking, and so has knowledge of the fate of a polyspermy-derived egg cell. In 2008 Rod Scott et al. made elegant use of *Arabidopsis thaliana tes* mutants, which form pollen with supernumerary sperm, to show that, similar to animal systems, polyspermy in the egg cell is typically prevented by an efficient polyspermy block (*Scott et al., 2008*). However, *Toda et al., 2016* were later able to artificially bypass the polyspermy

*For correspondence:
gross-hardt@uni-bremen.de

[†]These authors contributed equally to this work

Present address: [‡]School of Life Sciences, Westlake University, Hangzhou, Zhejiang, China; [§]Wageningen Plant Research, Bioscience, Wageningen, Netherlands

Competing interest: The authors declare that no competing interests exist.

block using delicate electrofusion experiments and combined two sperms with a single egg to generate polyploid plants.

In 2017, we provided molecular and genetic evidence of egg polyspermy in planta capitalizing on a high-throughput polyspermy detection (HIPOD) assay, which was established in the Landsberg *erecta* accession (*Nakel et al., 2017*). The assay consists of two pollen donors, that contain the individual elements of the yeast GAL4-UAS two-component system. The elements control expression of a herbicide resistance conferring gene, that consequently only gets activated if both paternal constructs are combined in a single egg cell. This enables the generation and identification of plants with three parents, one mother and two fathers (*Nakel et al., 2017*). Using a modified HIPOD assay that allows detection of embryos from three parents based on seed color, we also showed that polyspermy can specifically affect the egg cell, such that DNA of the second father bypasses DNA control mechanisms of the endosperm (*Mao et al., 2020*). This has significant implications for plant breeding, where the introduction of beneficial traits, for example from heat or drought-adapted wild species, is often prevented by hybridization barriers of the endosperm.

Plants can also be exposed to an insufficient number of fertilization-competent sperm (*Aw et al., 2010*; *Cyprys et al., 2019*; *Mori et al., 2006*; *Ron et al., 2010*; *Takahashi et al., 2018*; *von Besser et al., 2006*) in which case they activate a fertilization recovery program: The mechanism is triggered when fertilization in one of the female gametes stalls at the level of plasmo - or karyogamy and includes a transient halt of second synergid disintegration and concomitant attraction of supernumerary pollen tubes (*Beale et al., 2012*; *Kasahara et al., 2012*; *Maruyama et al., 2013*). This recovery program allows for recruitment of further sperm transporting pollen tubes and hence constitutes an important back-up mechanism.

Various proteases have previously been associated with the cortical reaction in animals, a fertilization-induced exocytosis of secretary granules that mediates a biochemical renovation of the egg coat to prevent further sperm entry (*Burkart et al., 2012*; *Haley and Wessel, 1999*). Based on the analysis of more than 450,000 seedlings, we here and in a preprint (*Mao et al., 2022*) show that the aspartic endopeptidases ECS1 and ECS2 repress polyspermy. The defect is associated with a transient delay in synergid disintegration, supernumerary pollen tube attraction, and heterofertilization which are characteristic features of the fertilization recovery program (*Beale et al., 2012*; *Kasahara et al., 2012*; *Maruyama et al., 2013*). In fact, we show that a fraction of *ecs1 ecs2* double mutants exhibit defects in gamete fusion and segregate seeds lacking paternally contributed fluorescence marker either in embryo or endosperm indicative of single fertilization events. These results are in line with a recent report by *Jiang et al., 2022* concordantly showing that ECS1 and ECS2 are required for gamete attachment. Furthermore, we show that the double mutants segregate haploid plants, as later confirmed by *Zhang et al., 2023*. ECS1 and ECS2 have previously been demonstrated to prevent the attraction of supernumerary pollen tubes by cleaving the pollen tube attractant LURE1 (*Yu et al., 2021*), indicating that polytubey is prevented by at least two different functions of ECS1 and ECS2.

## Results and discussion

### The aspartic proteases ECS1 and ECS2 negatively regulate polyspermy and the formation of triparental offspring

ECS1 and ECS2 were identified in a candidate gene approach for factors regulating polyspermy: T-DNA insertion lines for egg cell-expressed proteases, that is, *ecs1*, *ecs2-1*, *ecs2-2* single mutants, and *ecs1 ecs2-1* double mutants were subjected to HIPOD by synchronously applying the pollen of two fathers onto the stigmatic surface. One father contained a construct providing constitutive expression of the yeast transcription factor mGAL4-VP16 while the second father was engineered to express a herbicide conferring YFP-tagged BAR gene under the GAL4 responsive promoter UAS (*Figure 1A*). The resulting seedlings were subjected to herbicide selection (*Figure 1B*) and resistant plants were inspected under the fluorescence microscope for YFP expression (*Figure 1C*). The fusion of one egg cell with two sperm cells is expected to yield triploid rather than diploid offspring, which was demonstrated by flow cytometry (*Figure 1D*). Finally, we confirmed the triparental status of the seedlings by PCR, which revealed inheritance of GAL4 and UAS from the two genetically distinct fathers (*Figure 1E*). For wild-type Columbia plants, we recovered 19 triparental seedlings out of 92,680 seeds, corresponding to a polyspermy frequency of 0.041%. This is significantly higher than

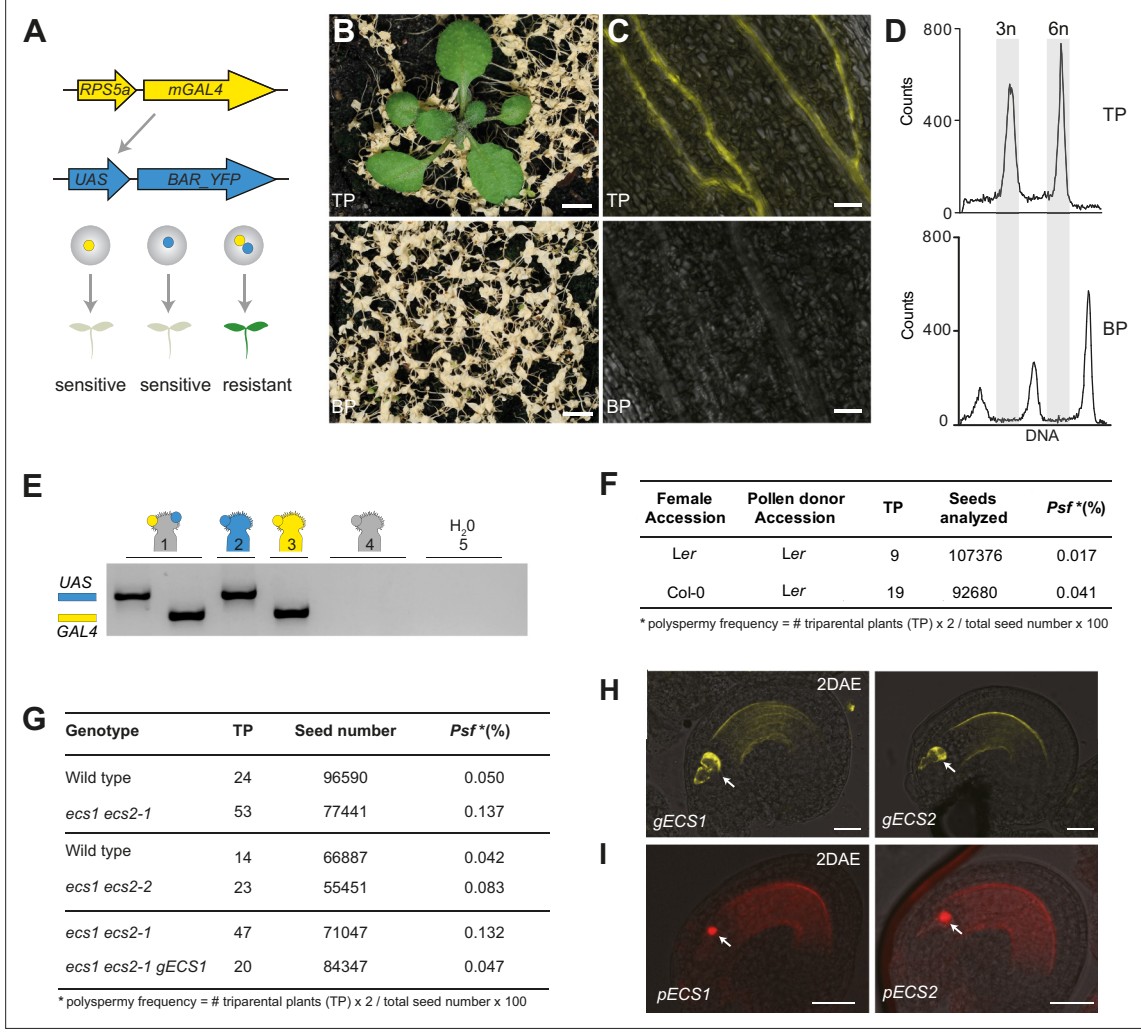

**Figure 1.** *ecs1 ecs2* double mutants segregate polyspermy-derived triparental offspring. (**A**) Illustration of the high-throughput polyspermy detection (HIPOD) assay. The two components of the yeast GAL4-UAS system are provided by distinct pollen donors (pollen donor 1, yellow and pollen donor 2, blue) such that herbicide resistance (green seedling) is only conferred if the mGAL4 transcription factor and the UAS-driven BAR_YFP protein are combined in a single egg (grey). (**B–D**) Herbicide treatment (**B**), YFP fluorescence (**C**), ploidy analysis (**D**) of triparental triploid plants (TP), and biparental diploid plants (BP). (**E**) PCR targeting *pUAS::BAR-YFP* (blue) and *pRPS5A::mGAL4-VP16* (yellow) in a herbicide-resistant plant recovered from HIPOD (1), *pUAS::BAR-YFP/+* (2), *pRPS5A::mGAL4-VP16/+* (3), wild-type control (4), water control (5). (**F**) Polyspermy frequency (Psf) following crosses of either wild-type L*er* (pollen acceptor) or wild-type Col-0 (pollen acceptor) with L*er* pollen donor 1 and L*er* pollen donor 2. (**G**) Polyspermy frequency in *ecs1 ecs2* mutants compared to wild type and a rescue line harboring *ecs1 ecs2 pECS1::gECS1-YFP*. (**H**) ECS1-YFP and ECS2-YFP localization before fertilization in the corresponding transgenic plants of *pECS1::ECS1-YFP* and *pECS2::ECS2-YFP*. Arrow indicates egg cell. (**I**) Promoter activity analysis using *NLS-tdTomato* expression driven by a promoter fragment upstream of either *ECS1* or *ECS2*. The arrow points to the egg cell nucleus. Scale bars, 5 mm (**B**), 50 µm (**C**) and (**I**), 20 µm (**H**).

The online version of this article includes the following source data and figure supplement(s) for figure 1:

**Source data 1.** Uncropped gel pictures for *Figure 1E*.

**Source data 2.** Raw data for polyspermy frequency in *Figure 1F and G*.

**Figure supplement 1.** Triparental plants recovered from *ecs1/- ecs2-1/+* heterozygotes predominantly segregate *ecs2-1*.

**Figure supplement 1—source data 1.** Uncropped gel pictures for *Figure 1A, B*.

the polyspermy frequency previously recovered for the Landsberg accession (*Nakel et al., 2017*), and experimentally re-assessed in this study (*Figure 1F*). By comparison, we detected an up to three-fold increase in polyspermy frequencies in *ecs1 ecs2-1* double mutants among the 77,441 seedlings analyzed (*Figure 1G*). ECS1 and ECS2 are hence the first molecular factors demonstrated to affect egg polyspermy and three parent formation in plants.

We confirmed that this phenomenon was causally linked to ECS1 by introducing a *pECS1::gECS1-YFP* rescue construct, which restored triparental frequencies to wild-type levels (*Figure 1G*). We next asked whether the polyspermy defect was of gametophytic origin. To answer this, we performed the polyspermy assay in *ecs1/ecs1, ecs2-1/ECS2* background, where each gynoecium is expected to segregate a 1:1 ratio of double and single mutant gametophytes. We first assessed *ecs2-1* segregation in the reference offspring population that was not herbicide treated by genotyping PCR and found an *ECS2:ecs2-1* ratio of 109:97, in line with the expected 1:1 segregation ratio. We furthermore reasoned that the segregation of the *ecs2-1* mutant allele would be enhanced in the triparental offspring if the absence of ECS2 in the female gametophyte positively affects polyspermy. Following herbicide selection of 78,338 seedlings, we recovered 24 triparental plants (*Figure 1—figure supplement 1A*), of which 18 plants exhibited an *ecs2-1* tDNA-specific PCR product (*Figure 1—figure supplement 1B*), corresponding to an *ECS2:ecs2-1* ratio of 6:18. This finding fits well with the threefold increase in polyspermy frequency found in *ecs1 ecs2-1* double mutants. In addition, analysis of YFP-tagged ECS1 and ECS2 fusion proteins, as well as *pECS1::NLS-tdTomato* and *pECS2::NLS-tdTomato* promoter reporter constructs revealed that ECS1 and ECS2 are exclusively expressed in the egg cell prior to fertilization (*Figure 1H and I*). Together, these results indicate a gametophytic origin of the polyspermy defect.

## *ecs1 ecs2* mutants are defective in gamete fusion and activate a fertilization recovery program

Making use of aniline blue staining, we found the reproductive defect to be associated with an 8.4% increase in supernumerary pollen tube attraction (*Figure 2A*). Polytubey in *ecs1 ecs2-1* double mutants was also reported by *Yu et al., 2021*, who provide a compelling set of experiments to show that the endopeptidases cleave specific variants of LURE1. In contrast to *Yu et al., 2021*, however, we observed reproductive defects, reflected by 13% abnormal seeds (*Figure 2B*). To investigate whether the phenotype was causally linked to ECS1 and ECS2, we first generated a second double mutant using the independent *ecs2-2* allele, which exhibited similar defects in seed development (*Figure 2B* and *Figure 2—figure supplement 1A*). Second, we introduced *pECS1:gECS1-YFP* and *pECS2:gECS2-YFP* into the *ecs1 ecs2* double mutants and demonstrated that each of the constructs restored fertility (*Figure 2B* and *Figure 2—figure supplement 1A*). Our results thus indicate that the seed defect in *ecs1 ecs2* double mutants is caused by loss of ECS1 and ECS2. To determine whether the defect was due to the role of ECS1 and ECS2 in the male or female reproductive tissue, we conducted reciprocal crosses between the double mutant and wild type. While the introduction of the mutant alleles from the female crossing partner recapitulated the defect, the reciprocal cross yielded fertile seeds, suggesting that the defect was of female origin (*Figure 2—figure supplement 1B*).

To further narrow down the reproductive defect, we performed a comprehensive analysis of cleared whole mounts of mutant and wild-type seeds at different developmental stages. Prior to fertilization, *ecs1 ecs2* female gametophytes exhibited a morphology similar to that of wild type (*Figure 2—figure supplement 1C*). Three days after pollination, however, we observed 8.1% of seeds that failed to develop an embryo and 7.1% of seeds that exhibited defects in endosperm development (*Figure 2C* and *Figure 2—figure supplement 1D*).

Together, the above-described results suggest a defect after or during fertilization. The latter scenario, that is, a defect in gamete fusion, has been shown to activate a fertilization recovery program, which delays synergid disintegration in order to attract additional pollen tubes (*Beale et al., 2012*; *Kasahara et al., 2012*; *Maruyama et al., 2013*). When comparing young *ecs1 ecs2* seeds with wild type, we indeed found that supernumerary pollen tube attraction correlates with a significant increase of synergid-containing seeds at the two-nucleate endosperm stage (*Figure 3A*). Overall progression of seed development between mutant and wild type was comparable (*Figure 3—figure supplement 1A*) and *ecs1 ecs2* synergid nuclei eventually degenerated (*Figure 3A*).

Our observations suggest a defect at the level of egg-sperm fusion in line with Jiang et al. However, they conflict with work by *Yu et al., 2021* and *Zhang et al., 2023* which reported that the defect manifests only during early embryo development. To gain a more detailed understanding of the mutant defect, we capitalized on an observation described by Takahashi et al. according to which the condensation status of sperm nuclei relaxes upon gamete fusion. Upon fertilization of *ecs1 ecs2* with HTR10-RFP (*Ingouff et al., 2007*) labeled sperm cells, we observed a pair containing two decondensed sperm nuclei signals in the wild type, indicating successful double fertilization (*Figure 3B* and

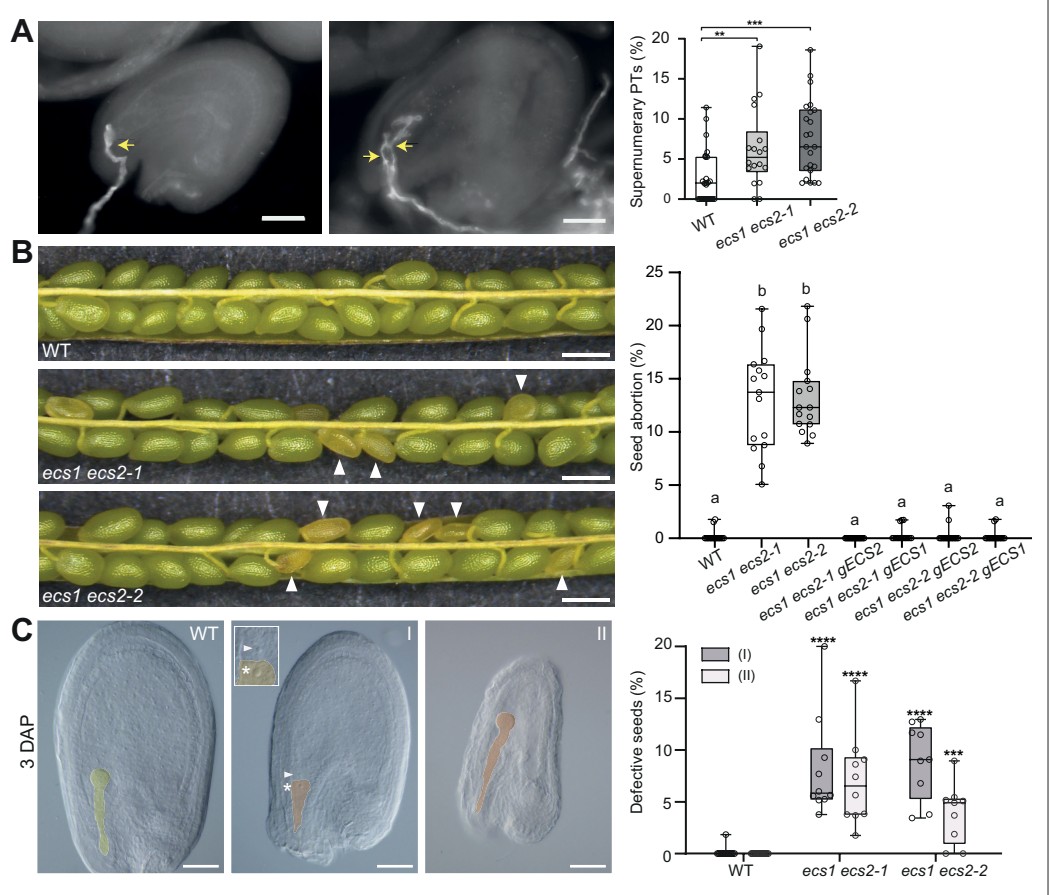

**Figure 2.** *ecs1 ecs2* double mutants exhibit reproductive defects. (**A**) Occurrence of polytubey in *ecs1 ecs2* mutants 20 hr after pollination (HAP) (n=1039/835/1070 for wild type/*ecs1 ecs2-1*/*ecs1 ecs2-2*, respectively). Circles represent data recovered from individual pistils. The images indicate one pollen tube (left) or two pollen tubes (right) targeting an embryo sac. Pollen tube is highlighted by a yellow arrow. (**B**) Silique of wild-type and *ecs1 ecs2* mutants with quantification of seed abortion in different genotypes (n>800 for each genotype). Circles represent data recovered from individual siliques. Arrowheads, aborted seeds. Different letters show significant difference; p<0.0001, F=106.0, by one-way ANOVA with a Tukey multiple comparison test. (**C**) Cleared whole mount and quantitative analysis of *ecs1 ecs2* mutant seed categories 3 days after pollination (DAP): In comparison to wild-type like seeds containing embryo and endosperm, *ecs1 ecs2* mutants segregate seeds without embryo (I) and seeds containing no or retarded endosperm (II) (n=655/540/522 for wild type/*ecs1 ecs2-1*/*ecs1 ecs2-2*, respectively). Circles represent data recovered from individual pistils. Embryo and unfertilized egg cells were false colored in yellow (wild type) and orange (mutant). Asterisk in image; unfertilized egg nucleus; arrowhead, endosperm nucleus. Similar results for (**B**) and (**C**) were obtained in independent experiments by a different scientist (data shown in *Figure 2—figure supplement 1A* and *Figure 2-figure supplement 1D*). Data in (**A–C**) are represented in box and whisker plot, center line, median; bottom and up, 25th and 75th percentiles. Whisker, the minimum and maximum. Two-tailed Mann-Whitney comparison test between wild type and mutants. Scale bars, 50 μm (**A**) and (**C**), 500 μm (**B**). **p<0.01; ***p<0.001; ****p<0.0001.

The online version of this article includes the following source data and figure supplement(s) for figure 2:

**Figure supplement 1.** The seed defect in *ecs1 ecs2* is of female origin.

**Figure supplement 1—source data 1.** Raw data for seed abortion in *Figure 2—figure supplement 1A*.

*Figure 3—figure supplement 2*). In the *ecs1 ecs2* mutants, we frequently detected a pair of sperm nuclei that differed in the condensation status analogous to the single fertilization pattern described by *Takahashi et al., 2018*. In one category the more dispersed signal corresponded to egg cell, (EC-F) in the other to central cell (CC-F) position indicating single fertilizations of either central (CC-F) or egg cell (EC-F) (*Takahashi et al., 2018*; *Figure 3B* and *Figure 3—figure supplement 2*). This is in line with

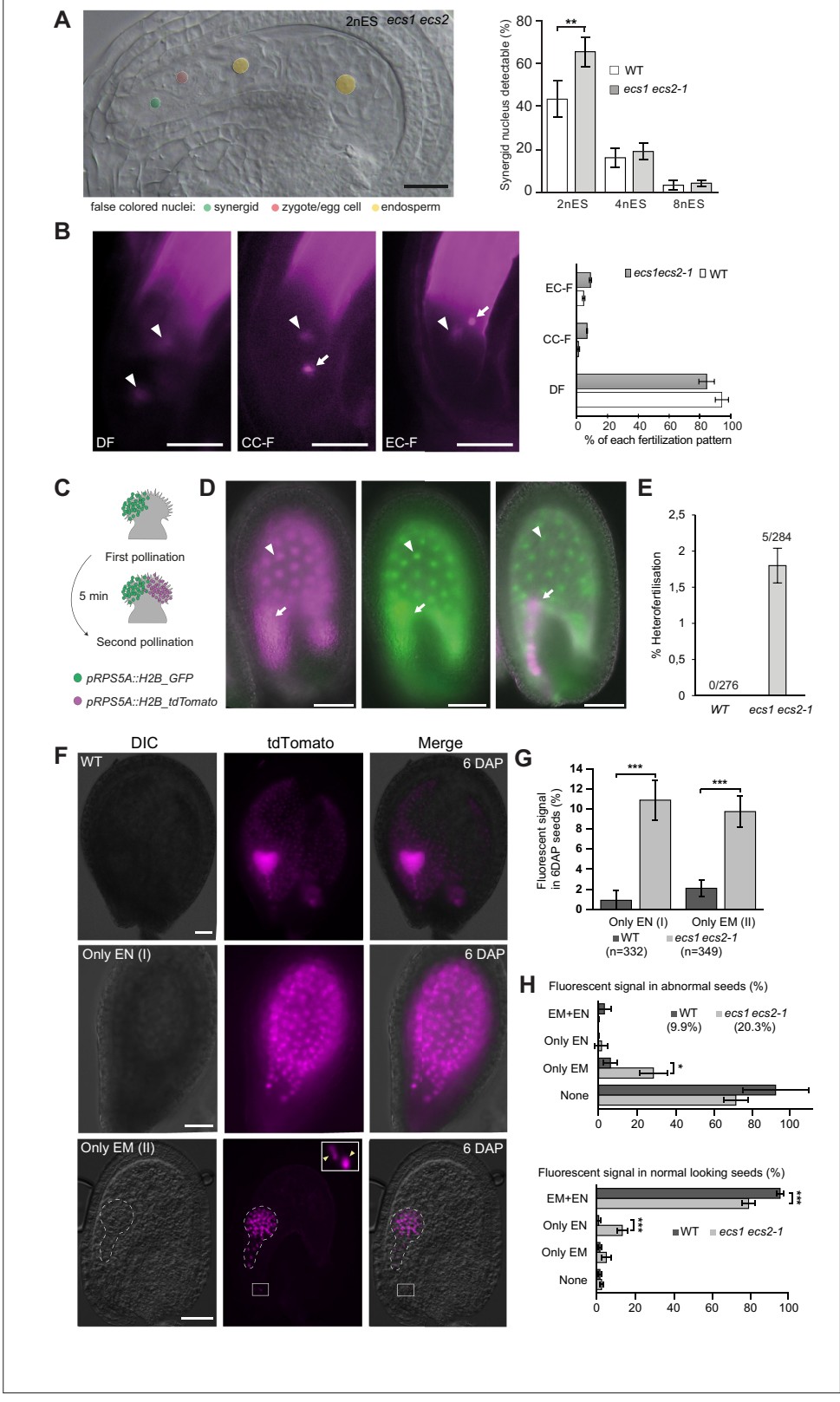

**Figure 3.** *ecs1 ecs2* double mutants segregate embryos that lack a paternal genome contribution. (**A**) Frequency of seeds containing a synergid nucleus at different endosperm stages (n for 2nES = 67/72, 4nES = 143/158, and 8nES = 179/141 for wild type/*ecs1 ecs2-1*, respectively). The image shows the second synergid nucleus at the two-nucleate endosperm stage. (**B**) Quantitative analyses of single fertilizations 8-10 HAP in *ecs1 ecs2* (n=90) and wild-

*Figure 3 continued on next page*

*Figure 3 continued*

type ovules (n=89) using HTR10-mRFP. Successful double fertilization (DF) is visualized by two dispersed sperm nuclei signals (arrow heads). In the middle and right panel, only one dispersed sperm cell nuclear signal (arrow heads) can be observed, while an additional sperm nuclear signal (arrows) is condensed suggesting the unfertilized state. The positions of the dispersed sperm signals correspond to the position of central cell (middle) and egg cell (right), respectively, and thus suggest different single fertilization patterns (CC-F and EC-F). (**C–E**) Analyses of seed development 3 days after dual pollination. (**C**) Diagram of the dual pollination experiment. (**D**) Successful double fertilization by a single pollen tube from either *pRPS5A::H2B_tdTomato* (left panel) or *pRPS5A::H2B_GFP* (middle panel). Heterofertilization involving the content of two genetically distinct pollen tubes is depicted in the right panel. Endosperm nuclear signal (arrow heads), embryo nuclear signal (arrows). (**E**) Percentage of heterofertilization events in wild type (n=276) and *ecs1 ecs2-1* (n=284) ovules. (**F**) Fluorescence signal in 6-day-old progenies of *ecs1ecs2-1* female plants upon paternal introduction of *pRPS5A::H2B-tdTomato*. WT: wild-type like seeds, showing paternal marker expression in both embryo and endosperm. Only EN (I): paternal marker expression only in endosperm. Only EM (II): paternal marker expression only in embryo. Dotted line; embryo, yellow arrowhead; sperm nuclei. (**G**) Percentage of 6-day-old progenies of *ecs1ecs2-1* or wild-type female plants crossed with *pRPS5A::H2B-tdTomato*, showing paternal marker expression in different fertilized tissues. (**H**) Assessment of paternally introduced marker expression in endosperm (EN) and embryo (EM) of 6-day-old siliques in *ecs1 ecs2-1* or wild type. The *ecs1 ecs2-1* seeds were divided into a normally looking and an abnormal seed fraction and analyzed individually. EM+EN; expression in both embryo and endosperm. (**G**) and (**H**), n=349/332 for wild type/*ecs1 ecs2-1*, respectively. Brightness was manually enhanced with Adobe Photoshop in (**A–C**). Data in (**A**), (**B**), (**G**), and (**H**) indicate mean ± SD. Two-tailed Mann-Whitney comparison test between wild type and mutants, *p<0.05; **p<0.01; ***p<0.001; ****p<0.0001. Scale bars, 20 µm (**A**), 25 µm (**B**), 75 µm (**D**), 50 µm (**F**).

The online version of this article includes the following figure supplement(s) for figure 3:

**Figure supplement 1.** Early seed development in *ecs1 ecs2* mutant.

**Figure supplement 2.** Detection of single fertilization events in *ecs1 ecs2* and wild-type ovules using HTR10-mRFP.

---

later observations using the same experimental setup, where we detect an increase of unfused sperm nuclei in *ecs1 ecs2* double mutants over wild type (***Figure 3—figure supplement 1B***).

In a complementary experiment we confirmed the fertilization defect and an active fertilization recovery program by genetic evidence: It was previously shown that egg and central cell can be fertilized by sperm from different pollen, making this so called heterofertilization a hallmark of single fertilization and an active recovery program (***Maruyama et al., 2013***): To visualize heterofertilization we pollinated *ecs1 ecs2* pistils with pollen from two distinct fathers carrying either *pRPS5A::H2B-GFP* or *pRPS5A::H2B-tdTomato* (***Figure 3C–E***). While we did not detect heterofertilization in wild-type (n=276), *ecs1 ecs2* mutants (n=284) exhibited a small fraction of seeds where the two fertilization products (egg and central cell) had inherited different fluorophores provided by the two different pollen donors (1.8%) (***Figure 3D–E***). This frequency is comparable to the one described by Maruyama et al., which observed heterofertilization using gamete fusion defective *kokopelli* sperm (***Maruyama et al., 2013***).

### *ecs1 ecs2* segregate maternal haploids

The early fertilization defect implies that the inheritance of paternal DNA is compromised in *ecs1 ecs2* mutants. To characterize this further we analyzed transmission of a paternally introduced *pRPS5a::H2B_tdTomato* marker (***Figure 3F***). While the majority of wild-type seeds showed a fluorescence signal in both endosperm and embryo, a significant number of *ecs1 ecs2* mutants contained seeds without paternal marker expression in the embryo (10.9%) or the endosperm (9.7%) (***Figure 3G***). When comparing marker distribution between aborted and normal seeds, we detected a relative enrichment of seeds without paternal contribution in the endosperm in the aborted seed fraction (***Figure 3H***), stressing the importance of a paternal endosperm contribution for seed development. Intriguingly, among the normal looking seeds we detected a fraction that lacked a paternal contribution in the embryo (***Figure 3H***).

This observation prompted us to trace the fate of such seeds. We inspected in total 200 *ecs1 ecs2* double mutant-derived plants and identified seven plants exhibiting short siliques and small flowers (***Figure 4A***). Flow cytometry revealed that all seven plants exhibited a haploid-characteristic signature, indicating that only one set of chromosomes had been inherited (***Figure 4B*** and ***Figure 4—figure***

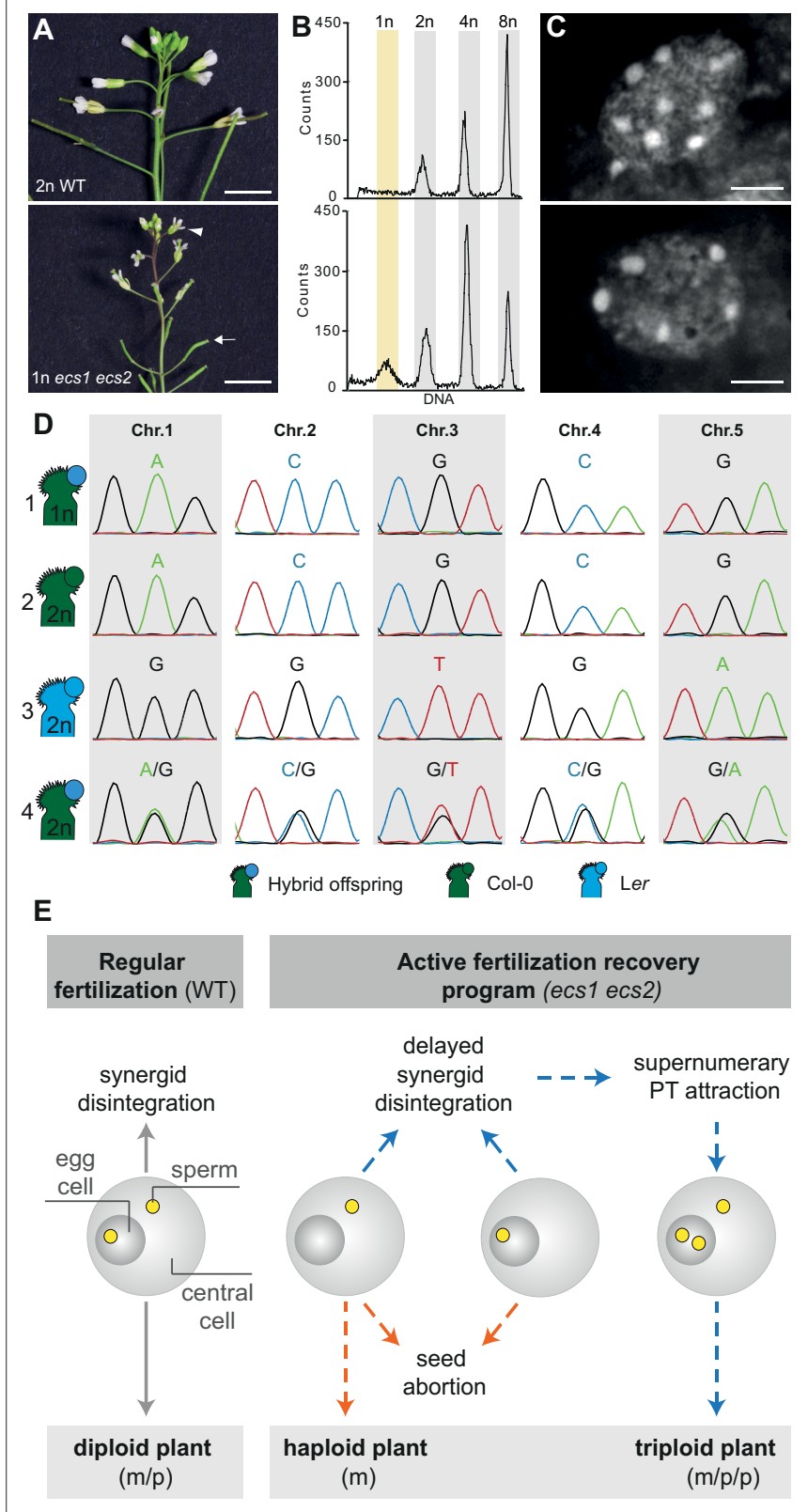

Figure 4. *ecs1 ecs2* double mutants segregate haploid offspring. (A–C) Inflorescence (**A**), flow cytometry (**B**), and DAPI stained chromosome spreads (**C**) of diploid (2n) wild-type and haploid (1n) *ecs1 ecs2* plant (n=200, see also *Figure 4—figure supplement 1*). Brightness was manually enhanced with Adobe Photoshop in (**A**). The arrowhead and arrow point at a small flower and an undeveloped silique, respectively. (**D**) Accession-dependent

*Figure 4 continued on next page*

*Figure 4 continued*

restriction fragment length polymorphisms (RFLPs) in the hybrid progeny from a cross of *ecs1 ecs2* mutant in Col-0 background (green) with L*er* wild type pollen (blue). Scale bars, 5 mm (**A**), 2 µm (**C**). (**E**) Schematic model illustrating causes and consequences of an active fertilization recovery program in *ecs1 ecs2*. Left panel: Double fertilization of both, egg and central cell, induces synergid disintegration in wild type. Right panel: Defects in double fertilization in *ecs1 ecs2* (egg cell or central cell only) result in delayed synergid disintegration and concomitant attraction of supernumerary pollen tubes increasing the likelihood of polyspermy (blue arrows). In addition, the fertilization defect contributes to seed abortion and can infrequently result in the generation of haploid plants (orange arrows). Dotted lines indicate incomplete penetrance of the respective phenotype. m and p indicate the segregation of maternal and paternal genomes, respectively. Note that according to *Yu et al., 2021* the attraction of supernumerary pollen tubes is additionally controlled by ECS1/2-meditated LURE1 cleavage after fertilization.

The online version of this article includes the following source data and figure supplement(s) for figure 4:

**Figure supplement 1.** Ploidy analysis of *ecs1 ecs2* double mutant offspring that deviates phenotypically from wild type.

**Figure supplement 2.** Inter-accession crosses between *ecs1 ecs2* (Col-0) and wild-type L*er* segregate low frequencies of haploid offspring.

**Figure supplement 3.** Haploid offspring derived from inter-accession crosses between *ecs1 ecs2* (Col-0) with wild type (L*er*) exhibit maternal genome signatures in restriction fragment length polymorphisms (RFLPs) analysis.

**Figure supplement 3—source data 1.** Uncropped gel pictures.

---

*supplement 1*). To substantiate this result, we performed a complementary approach and determined the chromosome number by chromosome spread. *A. thaliana* contains five different chromosomes and in the diploid Col-0 accession used in this study they exist in two copies. Notably, in nuclei of plants with short siliques and small flowers, we detected 5 instead of 10 chromosomes (*Figure 4C*), confirming the haploid nature of these plants. To determine whether these haploids were of maternal origin, we conducted inter-accession crosses combining diploid *ecs1 ecs2* in the Col-0 background with wild-type L*er* accession. We recovered eight haploid plants out of 934 F1 seedlings (*Figure 4— figure supplement 2*) and profiled accession-dependent restriction fragment length polymorphisms (RFLPs) for each of the five chromosomes. We only detected polymorphisms specific for Col-0 on all chromosomes in all eight haploids (*Figure 4D* and *Figure 4—figure supplement 3*), indicating that only the maternal genome had been inherited. Such maternal induction of haploidy has only been reported in a recent preprint for the gynoecium-expressed phospholipase AII (pPLAIIγ) (*Jang et al., 2022*) and these examples differ conceptually from earlier reported haploid inducers, which operate through selective elimination of aberrant paternal chromosomes (*Gilles et al., 2017*; *Karimi-Ashtiyani et al., 2015*; *Kelliher et al., 2017*; *Khanday et al., 2019*; *Li et al., 2021*; *Liu et al., 2017*; *Ravi and Chan, 2010*; *Underwood et al., 2022*; *Zhong et al., 2020*; *Zhong et al., 2019*).

The segregation of haploid plants in *ecs1 ecs2* mutants has similarly been observed by *Zhang et al., 2023*. However, in contrast to our work and the report by *Jiang et al., 2022*, Zhang et al. describe that the defect of *ecs1 ecs2* double mutants is triggered by semigamy post fertilization. According to previous work, semigamy scenarios would be expected to infrequently result in inheritance of paternal signatures (*Gerlach-Cruse, 1970*; *Lanaud, 1988*; *Rao and Narayana, 1980*) which was, thus far, neither observed by Zhang et al. nor us. Such a scenario would thus require the existence of an additional, previously undiscovered mechanism for the selective elimination of paternal wild-type DNA.

Several factors regulating gamete fusion have been described (*Aw et al., 2010*; *Cyprys et al., 2019*; *Mori et al., 2014*; *Mori et al., 2006*; *Rademacher and Sprunck, 2013*; *Ron et al., 2010*; *Sprunck et al., 2012*; *Takahashi et al., 2018*; *Tian et al., 2018*; *von Besser et al., 2006*). Among these, the sperm-specific AtDMP8 and AtDMP9 and the pollen-specific NLD have been implicated in haploid induction (*Gilles et al., 2017*; *Kelliher et al., 2017*; *Liu et al., 2017*; *Zhong et al., 2020*). It will be an interesting task for the future to determine whether other gamete fusion defective mutants similarly give rise to haploids or whether this phenotype uncovers an additional function of ECS1 and ECS2. In addition, it will be interesting to determine whether the activation of the fertilization recovery program is typically associated with increased polyspermy rates.

In summary, our and previous data suggest that ECS1 and ECS2 function at the level of gamete fusion (*Jiang and Qu, 2023*; *Jiang et al., 2022*; *Mao et al., 2022*) and support the coordination of

double fertilization. Thus, single fertilization of egg or central cell can be observed in *ecs1 ecs2* double mutants that contributes to seed abortion and can infrequently result in the generation of haploid plants. It also leads to delayed synergid disintegration and concomitant attraction of supernumerary pollen tubes. The supply of additional sperm cells can either lead to successful fertilization recovery or the induction of polyspermy (*Figure 4E*). It was recently reported that following successful double fertilization ECS1/2 are secreted and play a role in degradation of LURE peptides (*Yu et al., 2021*). To what extent this process in addition contributes to supernumerary pollen tube attraction is currently unclear.

Taken together, we have shown that ECS1 and ECS2 regulate offspring genome size via haploid and polyspermy induction, two agriculturally highly relevant mechanisms. This is particularly important, as the current pace of crop plant optimization is insufficient to meet future demand (*Hickey et al., 2019*). ECS1 and ECS2 provide molecular clues opening up new avenues for the introduction and/or combination of agricultural relevant traits.

## Materials and methods

### Plant material and growth conditions

All experiments were performed with *A. thaliana* L., ecotype Col-0 or L*er*. The seeds were sown on soil. Two days after stratification at 4°C, they were transferred and germinated in a Conviron MTPS growth chamber under long-day conditions (16 hr light/8 hr dark) at 23°C. Plants were transferred into 18°C after bolting. T-DNA insertion mutants SALK_021086 (*ecs1*), SALK_090795 (*ecs2-1*), SALK_036333 (*ecs2-2*) were obtained from European Arabidopsis Stock Center (NASC) (Nottingham, UK).

### Plasmid construction and plant transformation

The promoter and gene fragments of *ECS1* and *ECS2* were amplified from *Arabidopsis* Col-0 by using YM76s/as and YM78s/as and introduced into *pUAS::BASTA-YFP* (*Nakel et al., 2017*) using *AscI/NotI* restriction sites. The resulting constructs were then transformed into *ecs1 ecs2* double mutants by floral dip (*Zhang et al., 2006*). To generate *pECS1/2::NLS-tdTomato* transgenic plants, the promoter region of *ECS1* and *ECS2* was individually amplified from Col-0 by the respective primer pair YM76s/86as and YM78s/87as. The resulting fragment was then subcloned into DR13 plasmid (*pAt5g40260::NLS-tdTomato*) (*Völz et al., 2013*) through *AscI/PacI* restriction sites. Primer sequences used for genotyping and plasmid construction are listed in *Supplementary file 1*.

### Histology and microscopy

For the analysis of mature female gametophytes, the oldest closed flower buds were emasculated and harvested 2 days later. For the analysis of seed development, the emasculated flowers were manually pollinated and collected as specified in the figure legends. For whole-mount clearings, samples were treated with acetic acid/ethanol (vol/vol=1:9) overnight, washed in a 80%, 70%, and 50% ethanol series and mounted in chloral hydrate:glycerol:water (8:2:1; wt:vol:vol)(*Gross-Hardt et al., 2007*). For each experiment, more than five emasculated flowers or pollinated pistils were analyzed. With regard to the determination of persistent synergid cell, more than 10 pistils 19–21 hr after pollination (HAP) were scored and the data from the pistils containing less than 10 fertilized ovules were excluded. Cleared whole-mount ovules and seeds were analyzed under Zeiss Axioscope (Zeiss, Oberkochen, Germany) and the images were captured by Canon PowerShot G10 camera. Mature siliques were dissected and observed under Leica S6E or S8apo stereomicroscope (Leica, Germany).

For aniline blue staining, pistils were collected 20 HAP and processed as described previously (*Mori et al., 2006*). This experiment was analyzed for each genotype with two technical replicates. Supernumerary pollen tube attraction was examined using an epifluorescence inverted microscope (Leica DMI6000b) with DAPI filter.

### Triparental plant screening

To determine the occurrence of polyspermy-induced triparental plants, HIPOD screening was applied as previously described (*Nakel et al., 2017*). Two to three closed flower buds per inflorescence were emasculated. Two to three days after emasculation, pollen grains collected from plants of *pRPS5A::m-GAL4-VP16/+* (pollen donor 1, PD1) and *pUAS::BAR-YFP /+* (pollen donor 2, PD2) were applied onto

the stigmatic surface. Landsberg *erecta* accession was used for PD1 and PD2. The resulting mature seeds were sown on soil and the plants were subjected to herbicide treatment. The triparental status of herbicide-resistant plants was confirmed by flow cytometry, fluorescence microscopy inspection, and PCR targeting of the *GAL4* and *UAS* loci.

### Chromosome spread

To detect the exact chromosome number in the diploid and haploid plants, chromosome spreads from pollen mother cells were prepared according to published protocols (*Ross et al., 1996*).

### Ploidy analysis

Leaves of *ecs1 ecs2* plants or herbicide-resistant plants from HIPOD screenings were subjected to flow cytometry as previously described (*Nakel et al., 2017*).

### Statistical analysis

Data sets were processed in Microsoft Excel. Bar charts, box and whisker plots, as well as all statistical tests were generated using Microsoft Excel or GraphPad Prism.

## Acknowledgements

We thank Prof. Dr. Tetsuya Higashiyama (Nagoya University, Japan) for providing the pRPS5A::H2B-tdTomato and pRPS5A::H2B-GFP seeds. We thank Prof. Dr. Frederic Berger (Gregor Mendel Institute of Molecular Plant Biology, Austria) for providing transgenic plants expressing sperm marker pHTR10::HTR10-mRFP. We thank members of the Groß-Hardt laboratory for valuable comments on the manuscript. This work is supported by grants of the European Research Council (ERC Consolidator Grant 'bi-BLOCK' ID. 646644) and the European Innovation Council (EIC Transition '3P-Tec' 101057189) to RG.

## Additional information

### Funding

| Funder | Grant reference number | Author |
| --- | --- | --- |
| European Research Council | 646644 | Rita Groß-Hardt |
| European Innovation Council | 101057189 | Rita Groß-Hardt |

The funders had no role in study design, data collection and interpretation, or the decision to submit the work for publication.

### Author contributions

Yanbo Mao, Conceptualization, Investigation, Visualization, Methodology, Writing – original draft, Writing – review and editing; Thomas Nakel, Formal analysis, Investigation, Visualization, Methodology, Writing – review and editing; Isil Erbasol Serbes, Data curation, Formal analysis, Investigation, Visualization, Methodology, Writing – original draft, Writing – review and editing; Saurabh Joshi, Data curation, Investigation, Visualization, Writing – review and editing; Dawit G Tekleyohans, Formal analysis, Investigation, Visualization, Methodology, Writing – original draft, Writing – review and editing; Thomas Baum, Formal analysis, Investigation, Visualization; Rita Groß-Hardt, Conceptualization, Supervision, Funding acquisition, Project administration, Writing – review and editing, Writing – original draft

### Author ORCIDs

Yanbo Mao ![ORCID] https://orcid.org/0000-0001-5520-8202
Thomas Nakel ![ORCID] http://orcid.org/0000-0001-9033-5987
Isil Erbasol Serbes ![ORCID] http://orcid.org/0000-0001-7854-1243
Saurabh Joshi ![ORCID] http://orcid.org/0000-0002-4354-6762

Dawit G Tekleyohans (ID) https://orcid.org/0000-0001-7383-5971
Rita Groß-Hardt (ID) https://orcid.org/0000-0003-1998-0507

**Decision letter and Author response**
Decision letter https://doi.org/10.7554/eLife.85832.sa1
Author response https://doi.org/10.7554/eLife.85832.sa2

---

## Additional files

**Supplementary files**
- Supplementary file 1. Primer sequences used for genotyping and plasmid construction.
- MDAR checklist

### Data availability

All data generated or analysed during this study are included in the manuscript and supplementary files.

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
