## [Editor Report]

This important work convincingly shows that the endopeptidases ECS1 and ECS2 repress the formation of polyspermy-derived triparental offspring and haploid induction by promoting double fertilization. While the underlying mechanisms remain to be further elucidated, the data presented in this study represent a valuable foundation for understanding the regulation of offspring genome size. This study will be of particular interest to the large community of scientists who are interested in plant reproduction and breeding.

---

## [Decision Letter]

**Decision letter after peer review:**

Thank you for submitting your article "ECS1 and ECS2 suppress polyspermy and the formation of haploid plants by promoting double fertilization" for consideration by *eLife*. Your article has been reviewed by 3 peer reviewers, and the evaluation has been overseen by the Reviewing Editor, Hao Yu, and Jürgen Kleine-Vehn as the Senior Editor. The following individuals involved in the review of your submission have agreed to reveal their identity: Sharon A Kessler (Reviewer #1); Tomoko Igawa (Reviewer #2).

Essential revisions:

1. As the single fertilization events seem to be the most prominent phenotype in this study, additional evidence of single fertilization is necessary to determine whether the defects in ECS1 and ECS2 indeed influence gamete fusion.

2. This manuscript should be improved by providing clearer data analysis and descriptions (e.g. Figure 3), images with higher resolution or better representation (e.g. Figures 2B and 3B), and extra image data (e.g. the DIC image of the synergid nucleus in the embryo-only and endosperm-only ecs1 ecs2 ovules).

3. More discussions should be provided in the context of several recent studies reporting ECS1/2, and regarding the potential roles of ECS1/2 peptidases in regulating gamete fusion and/or repressing parthenogenesis.

Please also take into consideration the other specific comments from the reviewers below to revise the manuscript.

*Reviewer #1 (Recommendations for the authors):*

Very interesting story, but the presentation of the data in the text was a little confusing since it starts out with triploids, moves on to single fertilization, and ends with haploids but there is not a clear connection made for how all of these phenotypes relate to problems regulating gamete fusion. I'd suggest adding another paragraph at the end that brings these concepts together a little more clearly. It might be helpful to note that the single fertilization events seem to be the most prominent phenotype while the haploid induction and triparental triploids are quite rare. What does this tell us about the wild-type function of ECS1/2?

*Reviewer #2 (Recommendations for the authors):*

Although Zhang et al. have published similar data on haploid formation, the authors had released bioRxiv in advance, so I think there's no problem with novelty. The data quality is enough for a short report; however, the present descriptions are confusing and hard to understand soon. Please try to use uniform expressions in the description of each data. In addition, as I suggest in a public view, additional evidence of single fertilization would be critical to determine the ecs1ecs2 influence on gamete fusion. I encourage authors to try this or a similar alternative experiment.

*Reviewer #3 (Recommendations for the authors):*

1. The production of embryo-only seeds in ecs1 ecs2 mutants probably needs more explanation. They claimed that the embryo-only seeds could be observed in ecs1 ecs2 siliques. However, the embryo-only seeds in ecs1 ecs2 siliques are not reported in Jiang et al., (2022) and Zhang et al., (2023). Although the authors provided the reasons that this phenotype might result from the different growth conditions and developmental stages, the explanation is not satisfactory to me. I'm actually curious about one thing, that is, whether the percentage of the ovules with only embryo and arrested sperm cell nuclei is comparable with the percentage of embryo-only seeds (7.1%) in line 143. If they were comparable, the authors probably need to have a deeper discussion on how the egg cell- specifically expressed proteases only affect the interaction between the central cell and sperm cell in those embryo-only seeds. If they were not comparable, I probably have another explanation for these embryo-only seeds, that is, there is a possibility that both of the two sperm cells delivered by one pollen tube (in less pollinated condition) fertilize the egg cell simultaneously due to the polyspermy block defect in ecs1 ecs2 ovules. These ovules were not targeted by the second pollen tube, therefore the central cell did not fertilize in the end. Since the phenotype of having embryo-only seeds was completely not observed in the other two published papers, I would suggest the authors explain the cause of the embryo-only seeds more clearly.

2. In Figure 2B, the images of ecs1 ecs2 mutant siliques are not representative. The authors reported that there were two types of aborted seeds in siliques of ecs1 ecs2 mutants, including embryo-only seeds and endosperm-only seeds (Figure 2C). According to Xiong et al., (2021) and Li et al., (2022), embryo-only seeds do not enlarge and are small dot-like, which are significantly smaller than endosperm-only seeds and normal seeds. I notice that in Figure 2—figure supplement 1B, they showed small dot-like embryo-only seeds and white-but-smaller endosperm-only seeds in the siliques of ecs1 ecs2-1×ecs1 ecs2-1. Thus, I would suggest the authors provide representative images of siliques of ecs1 and ecs2 which simultaneously show the two types of aborted seeds and mark them respectively in Figure 2B.

3. In Figure 3—figure supplement 1B, I notice that the frequency of unfused sperm cells in ecs1 ecs2 ovules is approximately 15% (in my own calculation), but this ratio is significantly higher than the percentage of polytubey in ecs1 ecs2 mutants (Figure 2A). Therefore, I think that the phenotype of unfused sperm cells can not be fully attributed to and explained by the mechanism that is reported by Yu et al., (2021).

4. The logic about how ECS1 and ECS2 participate in persistent synergid cell degeneration is a little bit confusing. The authors think that the defect of persistent synergid cell degeneration is caused by the defect of gamete fusion. However, in Figure 3A, they provided the persistent synergid cell nucleus in ovules undergoing successful double fertilization. The conclusion and the evidence are two different things. I suggest that the authors show ovules with gamete fusion defect and polytubey to support that persistent synergid cell degeneration is caused by the defect of gamete fusion. For example, provide the DIC image of the synergid nucleus in the embryo-only and endosperm-only ecs1 ecs2 ovules. I also suggest the authors discuss the relationship between ECS1/2 and EIN3/EIL1/EIN2 since the phenotype in figure 3A is very similar to those of ein3 eil1 and ein2 mutants. I'm very much curious about whether the EIN3-dependent ethylene-response cascade that is activated by fertilization of the egg cell is affected in ecs1 ecs2 ovules. Whether the affected cleavage of pollen tube attractant AtLURE1 is related to persistent synergid cell death/nucleus disintegration in ecs1 ecs2 ovules?

In Figure 3B, what do the "unfertilized ovules" indicate? (Based on my understanding, this type of seed has neither embryo nor endosperm). The authors should explain this type of ovule in detail here.

---

## [Author Response]

Essential revisions:1. As the single fertilization events seem to be the most prominent phenotype in this study, additional evidence of single fertilization is necessary to determine whether the defects in ECS1 and ECS2 indeed influence gamete fusion.

We here provide two additional data sets that clearly show that *ecs1 ecs2* mutant plants indeed exhibit single fertilization, leading to fertilization recovery.

First, we used HTR10_RFP marker as described in Takahashi et al. and suggested by Reviewer 2 to analyze the decondensation of sperm cell nuclei 8-10 HAP. We indeed found an increase in single fertilization events as evidenced by condensed sperm cells either at the micropylar position (egg cell) or at the chalazal position (central cell) (Figure 3B).

Second, we performed a dual pollination experiment using *pRPS5A:H2B_GFP* and *pRPS5A:H2B_tdTomato* as male pollen donors and analyzed the fluorescence signal in the seeds 3-6 DAP. In *ecs1 ecs2* mutants we found 5 seeds that exhibited differential paternal contribution in embryo and endosperm, while we did not encounter a single such heterofertilization event in wild type. This experiment thus gives additional evidence, that after single fertilization of either egg or central cell, the fertilization recovery mechanism was activated and a second pollen tube (different color) was attracted and fertilized the hitherto unfertilized gamete. Please note that we found fertilization recovery at a frequency of 1.8 %, which is even slightly higher than the frequency reported by Maruyama et al. 2013 when using pollen tubes that contain a single sperm cell (1.2 %).

2. This manuscript should be improved by providing clearer data analysis and descriptions (e.g. Figure 3), images with higher resolution or better representation (e.g. Figures 2B and 3B), and extra image data (e.g. the DIC image of the synergid nucleus in the embryo-only and endosperm-only ecs1 ecs2 ovules).

Thank you for your suggestion. We added images for the silique analysis (Figure 1—figure supplement 1A) and now provide evidence for the different categories as suggested by Reviewer 2. In addition, we provide high resolution pictures throughout the manuscript for all figures. Finally, and in response to the comments by Reviewer 1, we exchanged the original Figure 3B with newly generated Figure 3B-E, to determine fertilization failure by checking the decondensation HTR10-RFP labelled sperm nuclei 8-10 HAP and the frequency of heterofertilization (see above).

3. More discussions should be provided in the context of several recent studies reporting ECS1/2, and regarding the potential roles of ECS1/2 peptidases in regulating gamete fusion and/or repressing parthenogenesis.

We added a model figure (Figure 4E) and included additional discussion to the manuscript: Line 207 to 210 and Line 223 to 232

Reviewer #1 (Recommendations for the authors):Very interesting story, but the presentation of the data in the text was a little confusing since it starts out with triploids, moves on to single fertilization, and ends with haploids but there is not a clear connection made for how all of these phenotypes relate to problems regulating gamete fusion. I'd suggest adding another paragraph at the end that brings these concepts together a little more clearly. It might be helpful to note that the single fertilization events seem to be the most prominent phenotype while the haploid induction and triparental triploids are quite rare. What does this tell us about the wild-type function of ECS1/2?

Thank you for your suggestions. We created a model figure (Figure 4E) to facilitate the understanding. In addition, we also added a paragraph in the end that summarizes the results and brings the concepts clearer together (Line 223 to 232)

Reviewer #2 (Recommendations for the authors):Although Zhang et al. have published similar data on haploid formation, the authors had released bioRxiv in advance, so I think there's no problem with novelty. The data quality is enough for a short report; however, the present descriptions are confusing and hard to understand soon. Please try to use uniform expressions in the description of each data. In addition, as I suggest in a public view, additional evidence of single fertilization would be critical to determine the ecs1ecs2 influence on gamete fusion. I encourage authors to try this or a similar alternative experiment.

Thank you for your suggestion. We rephrased the descriptions in the figure legends to make them easier to understand. In addition, we added the experiments suggested by the reviewer (see answers above).

Reviewer #3 (Recommendations for the authors):1. The production of embryo-only seeds in ecs1 ecs2 mutants probably needs more explanation. They claimed that the embryo-only seeds could be observed in ecs1 ecs2 siliques. However, the embryo-only seeds in ecs1 ecs2 siliques are not reported in Jiang et al., (2022) and Zhang et al., (2023). Although the authors provided the reasons that this phenotype might result from the different growth conditions and developmental stages, the explanation is not satisfactory to me. I'm actually curious about one thing, that is, whether the percentage of the ovules with only embryo and arrested sperm cell nuclei is comparable with the percentage of embryo-only seeds (7.1%) in line 143. If they were comparable, the authors probably need to have a deeper discussion on how the egg cell- specifically expressed proteases only affect the interaction between the central cell and sperm cell in those embryo-only seeds. If they were not comparable, I probably have another explanation for these embryo-only seeds, that is, there is a possibility that both of the two sperm cells delivered by one pollen tube (in less pollinated condition) fertilize the egg cell simultaneously due to the polyspermy block defect in ecs1 ecs2 ovules. These ovules were not targeted by the second pollen tube, therefore the central cell did not fertilize in the end. Since the phenotype of having embryo-only seeds was completely not observed in the other two published papers, I would suggest the authors explain the cause of the embryo-only seeds more clearly.

Thank you for your suggestion. We cannot exclude the possibility that the first pollen tube delivers both sperm cells to the egg cell. However, we now provide an extra data set that shows that we do encounter single egg cell fertilization (EC only– Figure 3B).

While secretion of ECS1 could not be observed (Susaki et al., 2021) before fertilization, it is not unlikely that these molecules are secreted directly at the initiation of fertilization and thus likewise influence the correct fertilization of the central cell. In addition it was previously shown that aspartyl proteases of the same class contain small peptides that can be cleaved off and act as mobile signal transducers (Xia et al., 2004).

2. In Figure 2B, the images of ecs1 ecs2 mutant siliques are not representative. The authors reported that there were two types of aborted seeds in siliques of ecs1 ecs2 mutants, including embryo-only seeds and endosperm-only seeds (Figure 2C). According to Xiong et al., (2021) and Li et al., (2022), embryo-only seeds do not enlarge and are small dot-like, which are significantly smaller than endosperm-only seeds and normal seeds. I notice that in Figure 2—figure supplement 1B, they showed small dot-like embryo-only seeds and white-but-smaller endosperm-only seeds in the siliques of ecs1 ecs2-1×ecs1 ecs2-1. Thus, I would suggest the authors provide representative images of siliques of ecs1 and ecs2 which simultaneously show the two types of aborted seeds and mark them respectively in Figure 2B.

We added Figure 2—figure supplement 1 containing also the class of embryo only seeds.

3. In Figure 3—figure supplement 1B, I notice that the frequency of unfused sperm cells in ecs1 ecs2 ovules is approximately 15% (in my own calculation), but this ratio is significantly higher than the percentage of polytubey in ecs1 ecs2 mutants (Figure 2A). Therefore, I think that the phenotype of unfused sperm cells can not be fully attributed to and explained by the mechanism that is reported by Yu et al., (2021).

Thank you for your observation. We agree with the reviewer’s assessment. We, however, believe that it is difficult to directly compare two differently conducted experiments using two different methods. We determined the fertilization failure by checking the decondensation HTR10-RFP labelled sperm nuclei 8-10 HAP (Figure 3B) and the frequency of heterofertilization through dual pollination experiment (Figure 3C-E) see above.

4. The logic about how ECS1 and ECS2 participate in persistent synergid cell degeneration is a little bit confusing. The authors think that the defect of persistent synergid cell degeneration is caused by the defect of gamete fusion. However, in Figure 3A, they provided the persistent synergid cell nucleus in ovules undergoing successful double fertilization. The conclusion and the evidence are two different things. I suggest that the authors show ovules with gamete fusion defect and polytubey to support that persistent synergid cell degeneration is caused by the defect of gamete fusion. For example, provide the DIC image of the synergid nucleus in the embryo-only and endosperm-only ecs1 ecs2 ovules.

Thank you very much for your comment. In the context of this particular experimental setup, we can only deduce that fertilization has occurred based on the division of the endosperm nuclei. At the same time, at this early stage the oocyte/zygote has not yet divided and therefore we cannot confidently distinguish between double fertilization or single fertilization of the central cell. Analysis of later time points requires introgression of fluorescent reporters which is beyond the time frame of this work but will be possible subsequently. We are sorry for the confusion and changed the figure caption accordingly (Figure 3A). In addition, we have included two experiments that clearly show single fertilization events (Figure 3B and Figure 3C-E). For further explanation of new experiments see above.

I also suggest the authors discuss the relationship between ECS1/2 and EIN3/EIL1/EIN2 since the phenotype in figure 3A is very similar to those of ein3 eil1 and ein2 mutants. I'm very much curious about whether the EIN3-dependent ethylene-response cascade that is activated by fertilization of the egg cell is affected in ecs1 ecs2 ovules. Whether the affected cleavage of pollen tube attractant AtLURE1 is related to persistent synergid cell death/nucleus disintegration in ecs1 ecs2 ovules?

We thank the reviewer for this observation. It was actually established earlier that in light of the fertilization-compensation system, all defects that interfere with gamete fusion are expected to extend the life-span of the second synergid (Beale, Leydon, & Johnson, 2012; Volz, Heydlauff, Ripper, von Lyncker, & Gross-Hardt, 2013). At the same time, we think that the involvement of the ethylene response cascade exceeds the scope of this report.

In Figure 3B, what do the "unfertilized ovules" indicate? (Based on my understanding, this type of seed has neither embryo nor endosperm). The authors should explain this type of ovule in detail here.

We removed the figure and exchanged with new data (see above).